# The Range Contraction and Future Conservation of Green Peafowl (*Pavo muticus*) in China

Mingxiao Yan [1,2,†], Bojian Gu [3,†], Mingxia Zhang [1], Wei Wang [4], Rui-Chang Quan [1], Jiaqi Li [5,*] and Lin Wang [1,*]

1   Southeast Asia Biodiversity Research Institute, Chinese Academy of Sciences & Center for Integrative Conservation, Xishuangbanna Tropical Botanical Garden, Chinese Academy of Sciences, Mengla 666303, China; yanmingxiao19@gmail.com (M.Y.); zhangmingxia@xtbg.org.cn (M.Z.); quanrc@xtbg.ac.cn (R.-C.Q.)
2   University of Chinese Academy of Sciences, Beijing 100049, China
3   Ministry of Education Key Laboratory for Biodiversity Science and Ecological Engineering, Coastal Ecosystems Research Station of the Yangtze River Estuary, Institute of Biodiversity Science, School of Life Sciences, Fudan University, No. 2005 Songhu Rd., Shanghai 200438, China; 19110700133@fudan.edu.cn
4   State Environmental Protection Key Laboratory of Regional Eco-Process and Function Assessment, Chinese Research Academy of Environmental Sciences, Beijing 100012, China; wang.wei@craes.org.cn
5   Nanjing Institute of Environmental Science, Ministry of Ecology and Environment, Nanjing 210000, China
*   Correspondence: lijiaqihao@163.com (J.L.); wanglin@xtbg.ac.cn (L.W.)
†   These authors contributed equally.

**Abstract:** The Green Peafowl (*Pavo muticus*) is vulnerable to anthropogenic pressures and has undergone an extensive decline through much of its range in Southeast Asia. However, little is known about the changing distribution of Green Peafowl in China through historical periods. We described a 5000–6000 years distribution change of Green Peafowl in China by using historical archives. We examined the present distributions of Green Peafowl by using camera traps and transect surveys and predicted the suitable habitat to support future conservation planning for this species. Although Green Peafowl was once widely distributed across China, the species experienced a southward range retreat over the past 5000–6000 years and is now restricted to a small part of Yunnan. The results of prediction from maximum entropy modeling (MaxEnt) showed that the size of suitable habitat of Green Peafowl in Yunnan was 17,132 km$^2$. The suitable habitat concentrated in nine prefectures of Yunnan and Pu'er, Chuxiong, and Yuxi accounted for 48.64%, 27.39% and 15.83%, respectively. These results suggest that central Yunnan can cover most of the current larger and more contiguous populations of Green Peafowl in China and should be protected. Moreover, some areas in southern Yunnan, such as Xishuangbanna, can be a candidate for reestablishing populations, given that the species disappeared in this region less than 20 years ago and has a large remaining habitat.

**Keywords:** historical distribution; distribution prediction; suitable habitat; conservation; protected area; camera trap





## 1. Introduction

Anthropogenic disturbances have threatened species persistence [1]. Many species have experienced severe declines in distribution and the populations have been restricted to fragmented habitats or small protected areas [2]. Understanding long-term species distribution change and current distribution patterns is crucial to develop appropriate conservation strategies for threatened species [3].

The Green Peafowl (*Pavo muticus*) was once widely distributed across Southeast Asia's tropical and subtropical forests [4], including Java, Peninsular Malaysia, Thailand, Cambodia, Laos, Vietnam, Myanmar, Southeast Assam, Bangladesh, and China. The population of Green Peafowl, however, has dramatically decreased since the beginning of last century, making it a species of high conservation concern [5–8]. A recent study revealed that the current geographic range of the species in mainland Southeast Asia is less than

16% of the assumed historical range [9]. Hunting and habitat loss are thought to be the major factors driving the decline of Green Peafowl populations [10–12]. The Green Peafowl is currently listed as Endangered in IUCN's Red List [13] and is included in Appendix II of the Convention on International Trade in Endangered Species of Wild Fauna and Flora (CITES).

In China, some archeological evidence and historical archive records show that Green Peafowl was widely spread in China and was regarded as a symbol of auspiciousness because of its beautiful appearance [14,15]. Due to anthropogenic disturbance, such as over-exploitation (meat and feathers were collected for food, decoration, and medicine, etc.) and habitat destruction, Green Peafowl's distribution range and population has undergone a drastic reduction in past decades [16–19], yet the distribution of Green Peafowl in China for a long time in history is not clear. Fortunately, some Chinese ancient texts, documents, and official government archives provide useful records about fauna and flora, which enable mapping of long-term distribution change in Green Peafowl distribution [19–21].

At present, the Green Peafowl is recognized as a Critically Endangered species on the latest RedList of China's Vertebrates [22,23]. Although the Chinese government has issued strict conservation policies to protect rare species, the conservation status of Green Peafowl in China remains precarious. Currently, the species is only distributed in the Yunnan Province of Southwest China. A study in the 1990s used interviews and line transect surveys to describe the species distribution in Yunnan and found Green Peafowls to occur in small and isolated populations in central, western, and southern parts of the province [24]. Using the same method, a recent study reported a further retraction of Green Peafowl's range and population size in Yunnan over the past 20 years [25]. The current distribution of Green Peafowl in China is confined to extremely small and fragmented populations with an average of 3–5 individuals per flock [26]. As elsewhere, habitat loss and direct persecution (hunting, egg collection, poisoning, etc.) are the major threats to Green Peafowl in Yunnan [27,28]. Current distribution range of Green Peafowl in China is usually summarized at coarse scale, such as county level, which cannot meet future conservation planning for this species. The species distribution is not subject to human's administrative divisions [29]. Thus, mapping the suitable habitat at fine scale to identify the potential range is urgently needed for the protection of Green Peafowl. Camera-trap surveys can provide presence data of rare species at fine spatial scale and the data was usually applied in species distribution models, such as maximum entropy modeling (MaxEnt), to predict suitable habitat and develop conservation strategies [30,31].

In this study, we had three objectives: (1) to provide an overview of historical distribution of Green Peafowl in China by literature review and using antique as circumstantial evidence; (2) to investigate Green Peafowl current range at fine spatial scale, mainly by employing camera traps and field transect surveys, and finally, (3) to provide protection recommendations based on the distribution of suitable habitat, such as the establishment of protected areas in the appropriate locations to conserve the scattered remaining populations of Green Peafowl, or population reintroduction in places with suitable habitat remain.

## 2. Materials and Methods

### 2.1. Historical Review of Distribution for Green Peafowl

In China, historical resources such as historical documents, antiques, and ancient paintings and proses could be utilized for reconstructing long-term distribution dynamics for a species [3]. The correct identification of Green Peafowl in the historical records is possible with the aid of the historical resources because Green Peafowl was a well-known species in China, and it was unlikely to be confused with other pheasant species due to its large body size, long slender neck, and unique plumage. Although Indian Peafowl (*Pavo cristatus*) has similar appearance, they are physically different enough to be easily distinguished. Green Peafowl have scaly green feathers on the neck (unlike Indian's smooth blue), narrowly erect spike-like coronal tuft (bare-shafted fan in Indian), and a yellow patch

on the face (absent in Indian). These characteristics can be distinguished in historical records, and some ancient paintings could be used as evidence [32] (Figures S1 and S2). We created a map to depict the distribution of Green Peafowl in China across historic periods following the next steps:

We used "Green Peafowl" or "Peacock" as the keyword to search the published literature from Google Scholar and Web of Science. Chinese names of species such as 绿孔雀, 孔雀, 孔鸟, and 越鸟 were also used in searching the China Academic Journals Full-text Database (CNKI https://www.cnki.net/ accessed on 9 June 2020). The literature that mentioned the historical distribution of Green Peafowl and peacock in China had been retained, including Jia and Zhang [14], Wen and He [19], Wang [33], and Bai [34]. From the literature, we found more Chinese antique books which recorded the occurrence of Green Peafowl. In these antique books, the ancient names of localities for Green Peafowl's occurrence were usually corresponding to administrative units at that historical period. According to the descriptions of these localities in the historical records, we identified the ancient names by means of local chronicles, chorography, and the China Administrative Division Database (http://www.xzqh.org/html/ accessed on 20 June 2020). The actual distribution ranges (usually shown as polygons in most studies) of Green Peafowl in history could not been delineated due to great changes in environment, such as forest loss, over a long period of time, even if the ancient localities have been identified. Therefore, we used points to represent the administrative units where Green Peafowl was once recorded. To assist in the identification of Green Peafowl, we also searched ancient Chinese prose in open websites (https://ctext.org/zhs, https://www.gushiwen.org/ accessed on 12 July 2020) and antiques with peafowl patterns in the museums' websites (https://www.dpm.org.cn/Home.html, https://www.npm.gov.tw/ accessed on 12 July 2020). In total, we collected 65 historical records of Green Peafowl from 37 references (Table S1). These references together comprise most of the available historical records of Green Peafowl in China.

According to archeological discovery, so far, the earliest record of Green Peafowl in China can be traced back to 3000 to 4000 BC. Accordingly, we segmented the recorded time of Green Peafowl into three time periods: from 4000 BC to the first year AD, from 1 to 1000 AD, and from 1001 to 1949 AD. The historical records would be merged into one record if they have the same locality but different recorded times (e.g., 500 AD and 600 AD) within a single time period.

*2.2. Mapping Potential Habitat for Green Peafowl*

2.2.1. Field Surveys for Occurrence Data

Current distribution of Green Peafowl in China was limited to several prefectures in Yunnan province [25,26]. We investigated seven prefectures where Green Peafowl may occur, including Chuxiong, Yuxi, Dali, Dehong, Lincang, Pu'er, and Xishuangbanna in Yunnan. From 2013 to 2019, we conducted some interviews (with ornithologists, ecologists, forestry officials, forest rangers of nature reserves, members of non-governmental organizations, and villagers) to determine the approximate ranges where Green Peafowl once occurred. The questions for interviewees included the following: (1) Can you identify Green Peafowl by its appearance, feathers, footprints, or call? (2) Have you ever seen Green Peafowls (or feathers, footprints) or heard their calls in recent years? (3) Where and when did you see Green Peafowls or hear their calls? (4) Do you know anyone else who knows the distribution of Green Peafowl? After the interviews, camera-trap and transect surveys were implemented in reported ranges. Monitoring in these areas was conducted for at least three months with camera traps or was visited for transect surveys at least twice in the breeding season (December to June) since males call more during the breeding period [35,36].

A total of 321 camera-trap stations were stationed at 52 localities/points. The distance between two localities was more than 5 km. The number of camera-trap stations at each point ranged from 5 to 10, depending on the local topography. Infrared cameras (Ltl-

6511MC; Ltl Acorn, Zhuhai, China) were attached to trees (0.5–1.2 m above ground) along animal trails. The minimum distance between any two camera-trap stations was >500 m, and the time between two successive triggers was 5 s, each trigger took three consecutive images [37,38]. All cameras worked 24 h per day.

In addition to camera-trap surveys, we also conducted line transects to record the presence and absence of Green Peafowl in localities that could not be surveyed with camera traps. A total of 60 localities/points were surveyed with field transects. The length of each transect ranged from 0.5 to 10 km, depending on topographical features and vegetation area. The total length of the transect surveys was 224.2 km. The transects were conducted between 06:00–10:00 a.m. and 17:00–20:00 p.m., the times of peak Green Peafowl activity [39,40]. Green Peafowl's footprints, feathers, calls, and feces and their GPS coordinates were recorded on each transect. We used Garmin 62S (Garmin International, Inc., Olathe, KS, USA) to record the GPS coordinates. Occurrence data of Green Peafowl was collated from the camera-traps and the transect surveys (Table S2) and used it for species distribution modeling.

### 2.2.2. Environmental Variables

We used 22 environmental variables to predict the potential distribution area for Green Peafowl. Nineteen bioclimatic variables were obtained from the WorldClim database [41]. Land-cover data were derived from the European Space Agency (Globcover 2009, http://due.esrin.esa.int/page_globcover.php accessed on 16 July 2020) and was classified into 13 types [9]. Human pressures index was obtained from The Global Human Influence Index Dataset of the Last of the Wild Project, Version 2, 2005 [42]. The digital elevation model (SRTM, Shuttle Radar Topography Mission) data was downloaded from the SRTM 90 m Digital Elevation Database (http://srtm.csi.cgiar.org accessed on 16 July 2020).

### 2.2.3. Green Peafowl Distribution Modeling

We used maximum entropy modeling software (MaxEnt version 3.4.1) to predict the potential habitat of Green Peafowl as MaxEnt has been the most widely applied in species distribution modeling with the presence-only data [31,43]. Thirty-two records of Green Peafowl and twenty-two environmental variables were used in MaxEnt. Twenty-five percent of presence data of Green Peafowl was randomly selected for evaluating the model performance. An output ranging from 0 (unsuitable) to 1 (maximum suitability) was provided to present the probabilities of presence for Green Peafowl [44]. We used two logistic thresholds in MaxEnt '10 percentile training presence' and 'Balance training omission, predicted area, and threshold value' to determine the habitat level as the former is most accurate and closer to the real distribution [45], and the latter has the minimum omission rate and thus can model the species' potential distribution adequately [31,46]. Accordingly, habitat of Green Peafowl was divided into three levels: suitable habitat (≥10 percentile training presence), marginal habitat (between '10 percentile training presence' and 'Balance training omission, predicted area, and threshold value'), and unsuitable habitat (<Balance training omission, predicted area, and threshold value). Marginal habitat means the habitat which species may be able to persist in but at lower densities [47]. We predicted the potential habitat for Green Peafowl in the region of 20°0′–30°0′ N, 95°0′–119°0′ E (Figure 2b) since the region has similar climate and geography characteristics and covers all the current known distribution areas of Green Peafowl in China.

## 3. Results

### 3.1. Historical Distribution of Green Peafowl in China

The distribution of Green Peafowl from 5000–6000 (3000–4000 BC) years ago to the establishment of the People's Republic of China (1949 AD) was illustrated in Figure 1. A total of 68 records were identified (Table S1). The species once lived in around seven of China's provinces/autonomous regions. The earliest evidence of a Green Peafowl bone was buried in ashy layers in a heritage site in Xichuan county, Henan Province (red

spot with No. 1 in Figure 1a); it was estimated that the species occurred in this region around 5000–6000 years ago [14,16]. Three records were in central China (Shanxi and Hubei provinces) and southern China (Guangdong province) during the period of 1 to 1046 BC. Twenty-four records were in south and southwest China from 1 to 1000 AD. For the 37 records from the past millennium (1001–1949 AD), all were in south and southwest China (Figure 1a). Since the Qing dynasty (1636–1911 AD), the China's Green Peafowl population has been confined to Yunnan, Guangxi, and Guangdong (Figure 1b,c). In addition, three records were identified as Indian Peafowl (Table S1 and Figure S2).

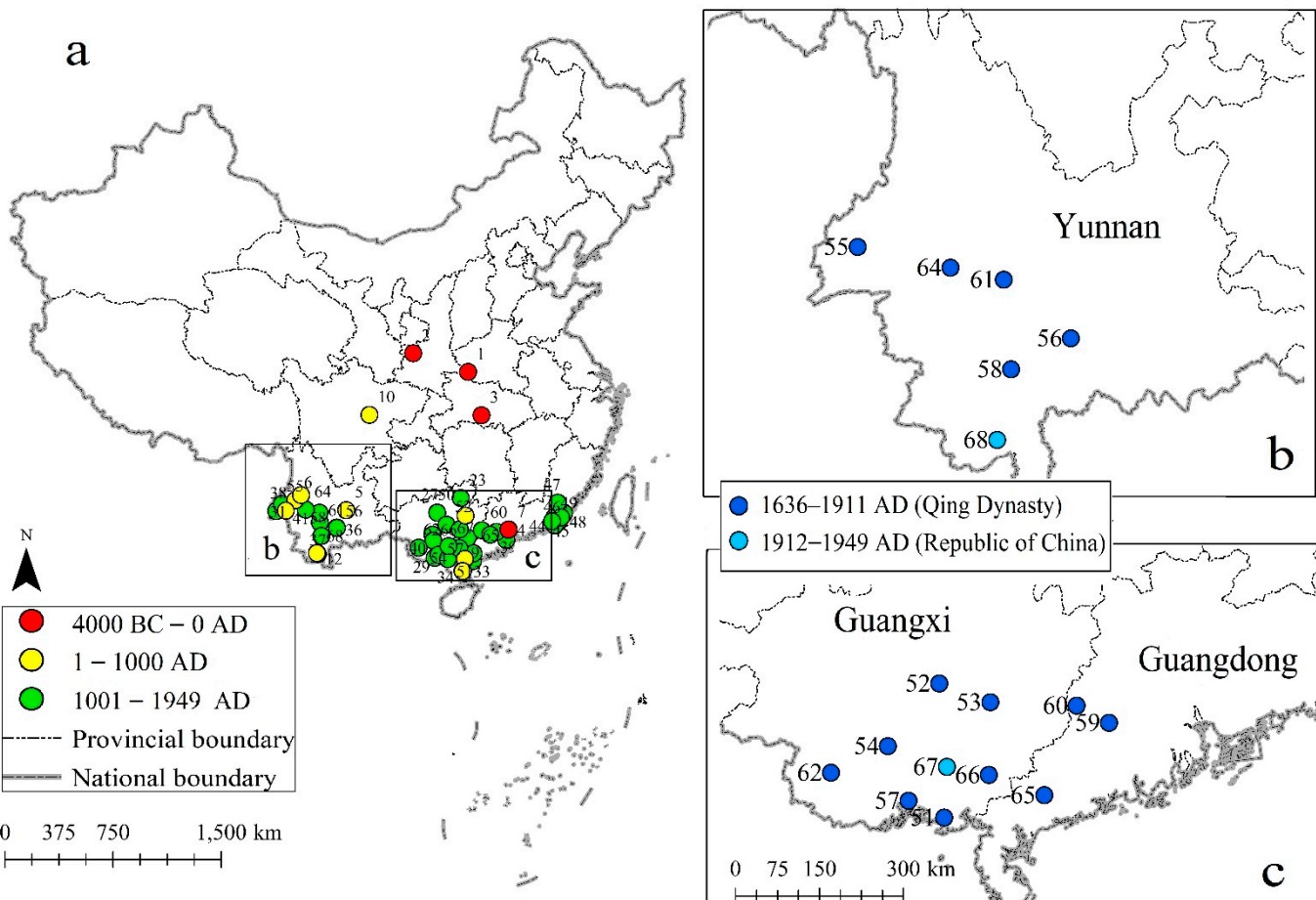

**Figure 1.** The historical records of Green Peafowl in China from 3000–4000 BC to 1949 AD (**a**). Green Peafowl was only documented in Yunnan Province (**b**), Guangxi Zhuang Autonomous Region, and Guangdong Province (**c**) during the Qing Dynasty (1636–1911 AD) and the Republic of China (1912–1949 AD).

### 3.2. Field Survey Results

From 2013 to 2019, we surveyed 52 localities with camera traps, accumulating 85,890 trap nights from 321 camera-trap stations. Green Peafowl were photo-captured in 27% (14 out of 52) of the localities. In that period, we also surveyed a total of 77 transects in 60 localities and we found evidence of Green Peafowl presence at 30% (18 out 60) of them (Table S2). In total, Green Peafowl appeared at 32 localities in Yunnan, of which three-quarters (24) of the distribution localities were concentrated in Yuxi City and Chuxiong Yi Autonomous Prefecture, both in central Yunnan (Figure 2a).



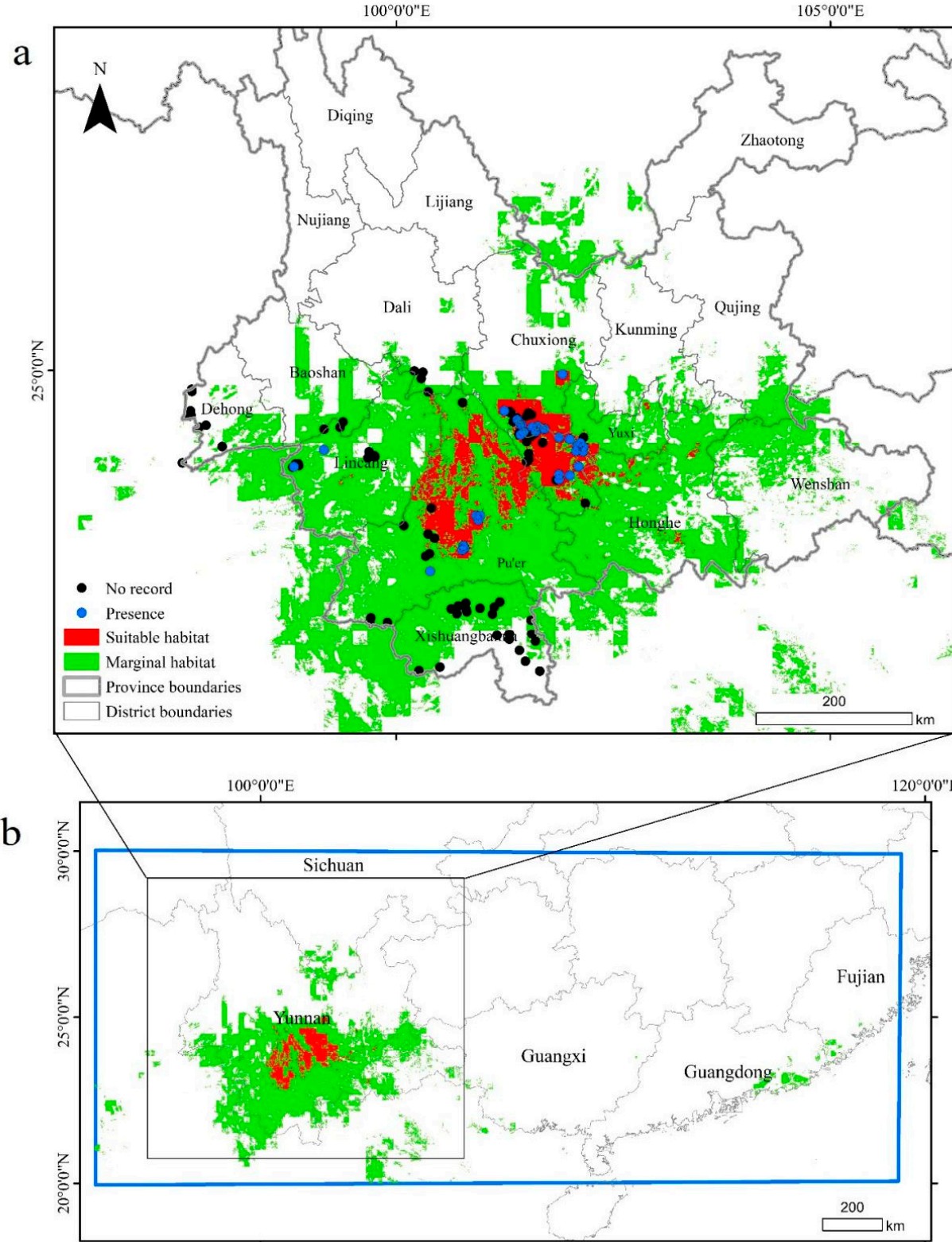

**Figure 2.** The suitable and marginal habitats of Green Peafowl in Yunnan Province (**a**); the dots represent the survey sites for the camera-traps and line transects. The blue rectangle in (**b**) presents the region of prediction of potential Green Peafowl habitat.

### 3.3. Potential Habitat of Green Peafowl

The value of the Area Under the Receiver-Operator Curve (AUC) was 0.993, which indicated that the model has good performance in the prediction of the potential habitat for Green Peafowl. Annual precipitation (Bio12) and the precipitation of the driest month (Bio14) made the most important contributions in our prediction model (Table S3). The suitable habitat (predicted probabilities of presence > 0.313) area of Green Peafowl is 17,132 km$^2$. All the suitable habitat was distributed in nine districts of Yunnan province (Table 1, Figure 2a), and Pu'er, Chuxiong, and Yuxi accounted for 48.64%, 27.39%, and 15.83%, respectively. The marginal habitat in China (predicted probabilities of presence between 0.011 and 0.313) was distributed in five provinces, including Yunnan, Guangxi, Guangdong, Sichuan, and Fujian with a total area of 142,681 km$^2$, and Yunnan accounted for 90.84% (Table 1, Figure 2b).

**Table 1.** Predicted suitable habitat and marginal habitat for Green Peafowl in China; the suitable habitat only distributed in Yunnan province.

| Province | District | Habitat Type | Area (km$^2$) | Percentage |
|---|---|---|---|---|
| Fujian | | Marginal | 1141 | 0.80% |
| Guangdong | | Marginal | 4614 | 3.23% |
| Guangxi | | Marginal | 2500 | 1.75% |
| Sichuan | | Marginal | 4821 | 3.38% |
| Yunnan | | Marginal | 129,605 | 90.84% |
| Yunnan | | Suitable | 17,132 | |
| | Pu'er | Suitable | 8333 | 48.64% |
| | Yuxi | Suitable | 4693 | 27.39% |
| | Chuxiong | Suitable | 2712 | 15.83% |
| | Honghe | Suitable | 880 | 5.14% |
| Yunnan | Lincang | Suitable | 449 | 2.62% |
| | Wenshan | Suitable | 28 | 0.16% |
| | Dali | Suitable | 22 | 0.13% |
| | Baoshan | Suitable | 12 | 0.07% |
| | Qujing | Suitable | 3 | 0.02% |

## 4. Discussion

By means of historical resources, such as historical documents, antiques, and ancient paintings and proses, we mapped the distribution change for Green Peafowl in China, which covered a period from 5000–6000 years ago to 1949 AD. Green Peafowl were once widely distributed through much of China [16,18,19], which may reflect why the species is such a well-known art symbol for Chinese people [32,48]. In addition to Green Peafowl, the historical resources also provided effect records for some species in China, such as the Asian elephant (*Elephas maximus*), Brown-eared pheasant (*Crossoptilon mantchuricum*), and snub-nosed monkeys (*Rhinopithecus*), and help to reconstruct long-term distribution dynamics [3,20,49]. In our study, one record from Hami, Xinjiang Uyghur Autonomous Region, northwestern China, was identified as Indian Peafowl according to a painting from Qing Dynasty which clearly depicted the characteristics of Indian Peafowl (Table S1 and Figure S2). Two other records (Hetian and Akeshu, Table S1) also from the Xinjiang Uygur Autonomous Region were also identified as Indian Peafowl since the three localities were the major towns on the ancient Silk Road. It is more likely that Indian Peafowls from the three localities were commodities or pets brought to China from ancient India via the Silk Road rather than the result of natural dispersal. So far, there have been no reports about the wild populations of Indian Peafowl in China.

Green Peafowl was distributed widely in southwest (Yunnan Province) and southern China (Guangxi Zhuang Autonomous Region and Guangdong Province) even in the Qing dynasty (1636–1911 AD, Figure 1b,c). The reduction in the species distribution in southern China occurred from the later Qing dynasty due to immigration and human population



growth in southern China and war-torn territories [50]. After that, there is only one record of Green Peafowl from these regions in the Republic of China (1912–1949 AD) (Figure 1c). After 1949 AD, a reference mentioned that Green Peafowl might distributed in southeast Tibet [51], but a recent survey from Kong et al. [26] with interview and line transects showed that they did not find Green Peafowl in Motuo and Chayu in southeast Tibet. Now, the extant Green Peafowl populations in China are more likely to be confined to fragmented areas only within Yunnan province. The estimated population decreased to a quarter (240–280 individuals [25,26]) compared to the 1990s (1100 individuals, [24]), and the ranges also retracted to two-thirds of the assumed range of 20–30 years ago [26]. Different kinds of persecution such as hunting, trade, and habitat degradation are the most important reasons for range retraction and population decline of Green Peafowl in historical times in China. China has a long history of anthropogenic impacts on the survival and distribution of fauna and flora. Even the most remote regions were also inevitably impacted, not only Green Peafowl. Other species in the region, for example, the snub-nosed monkeys [49], giant pandas (*Ailuropoda melanoleuca*), Asia elephants, rhinoceroses (*Dicerorhinus sumatrensis*) [52], and gibbons (Hylobatidae) [21], have all experienced population declines and range contractions as a consequence of anthropogenic pressure in the recent centuries.

Comparing with previous studies which provided Green Peafowl's distribution information in Yunnan based mainly on county level [24–28,53,54], our field surveys were implemented at a much finer scale. The camera traps combined with the line transects to detect the presence of Green Peafowl, which supported the prediction of potential habitat for Green Peafowl. In our exhaustive field surveys, we did not find evidence of Green Peafowl presence in areas where they are supposed to occur, such as in the counties of Jinghong, Longchuan, Longling, Ruili, and Changning [26]. Remaining populations are still threatened with extirpation if appropriate conservation measures are not employed optimally and timely [55].

From the predicted results, central Yunnan (Pu'er, Yuxi, and Chuxiong) held most of the remaining suitable habitat of Green Peafowl in China, accounting for 91.86% of the suitable habitat. Central Yunnan has the highest priority to be protected to include most the small-fragmented populations in this region. In addition to Green Peafowl, our camera-trap surveys also found other rare species of conservation concern occurring in these areas, such as forest musk deer (*Moschus berezovskii*), rhesus macaque (*Macaca mulatta*), Mrs Hume's Pheasant (*Syrmaticus humiae*), Lady Amherst's Pheasant (*Chrysolophus amherstiae*), and other animals with National Priority Level I or II of Protection, which also will benefit from the establishment of protected areas.

Reintroduction, as in the cases of crested ibis (*Nipponia nippon*) in China [56] and white-handed gibbon (*Hylobates lar*) in Thailand [57], may be a good option to save the species in China. Southern Yunnan such as Xishuangbanna has the potential to re-establish populations, given that the species disappeared in this region less than 20 years ago and have a large remaining habitat (Figure 2a). Some places in Yunnan where small populations of Green Peafowl are still present (blue spots in Figure 2a) have the priority to promote habitat connectivity via appropriate restoration [58]. Despite the opportunities such as the establishment of a larger protected area and population reintroduction for saving Green Peafowl in China, improving the management and law enforcement in the existing protected areas to stop or decrease poaching, poisoning, and egg collection are particularly important to the population viability.

## 5. Conclusions and Suggestions

In conclusion, this study shows the distribution shrinkage of Green Peafowl in China over an extended period. Green Peafowl in China have now been confined in fragmented areas within the Yunnan province. If we do not take the necessary steps to conserve this species, it will undoubtedly disappear soon in China. To reverse this trend, emergent actions are suggested to be taken, including reducing persecution, increasing population size, steadily expanding optimal habitats, as well as population re-establishment in areas of the

previous distribution range. Central Yunnan holds 91.86% of the suitable habitat for Green Peafowl in China, thus we recommend that a larger protected area can be established there to promote connectivity among the scattered localities where Green Peafowls distribute. However, the boundary of such a protected area has not been delineated in this study due to a lack of accurate and up-to-date maps such as villages and roads. Further analyses should also be conducted by integrating appeals of stakeholders (government, enterprises, local communities, and scientists) into conservation planning.

**Supplementary Materials:** The following are available online at https://www.mdpi.com/article/10.3390/su132111723/s1, Figure S1: Green Peafowl and Indian Peafowl are physically different enough to be easily distinguished, and Green Peafowl was a prototype in the artwork creations of ancient China. Figure S2: The distribution ranges from IUCN for Green Peafowl and Indian Peafowl and a painting for Emperor Qianlong as a tribute which painted two Indian Peafowls. Table S1: Collected records for mapping the historical distribution of Green Peafowl in China. Table S2: Camera traps and line transects for acquiring presence data of Green Peafowl. Table S3: Percent contribution and permutation importance for environmental variables in the prediction of potential habitat for Green Peafowl.

**Author Contributions:** M.Y.: Conceptualization (equal); formal analysis (lead); investigation (equal); methodology (equal); writing—original draft (lead). B.G.: data curation (lead); investigation (equal); methodology (equal); resources (equal); writing—original draft (supporting). M.Z.: investigation (equal); writing—original draft (supporting). W.W.: resources (equal); writing—original draft (supporting). R.-C.Q.: conceptualization (equal); funding acquisition (equal); writing—review and editing (supporting). J.L.: conceptualization (equal); funding acquisition (equal); resources (equal). L.W.: conceptualization (equal); project administration (lead); funding acquisition (equal); writing—review and editing (lead). All authors have read and agreed to the published version of the manuscript.

**Funding:** This research was funded by the National Natural Science Foundation of China (31872963); the Biodiversity Investigation and Assessment Program (2019–2023) of Ministry of Ecology and Environment of China; the Lancang-Mekong Cooperation Special Fund (Biodiversity Monitoring and Network Construction along the Lancang-Mekong River Basin project); the CAS 135 program (No. 2017 XTBG-F03); and Transboundary Biodiversity Conservation in the Gaoligong Mountain and International Cooperation System Construction.

**Institutional Review Board Statement:** Not applicable.

**Informed Consent Statement:** Not applicable.

**Data Availability Statement:** The data of historical distribution for Green Peafowl in China and the occurrence data of Green Peafowl for the prediction of potential habitat are available in Supplementary Materials.

**Acknowledgments:** We thank the Chinese Felid Conservation Alliance, Wild China Film, Friends of Nature and Cloud Mountain Conservation for offering clues and help. We appreciate Lianxian Han from the Southwest Forestry University and Cheng Wen and Shishun Zhou from the Xishuangbanna Tropical Botanical Garden for the suggestions to the field work. We are grateful for the permissions and field assistance of these governmental managers and staffs, including Yingcai Ni, Kaichun Xiong, and other staffs from the Yubaiding Nature Reserve; Jiacang Zhao from the Tongbiguan Provincial Nature Reserve; Shengxi Yang and Jian Wang from the Nuofu Forestry management; the Qinghua Provincial Nature Reserve, Daxueshan National Nature Reserve, and Jingdong Subtropical Botanical Garden and the Yuanjiang National Nature reserve. We also thank Guogang Li, Liping Zhou, Kang Luo, Xiaobao Deng, Ruinian Li, Linzhuang Bai for the helps in the field work. We also appreciate Eben Goodale from the Guangxi University and Ahimsa Campos Arceiz and Alice C. Hughes from the Xishuangbanna Tropical Botanical Garden for their helps in writing this paper.

**Conflicts of Interest:** The authors declare no conflict of interest.

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
