# Peer review of "The Range Contraction and Future Conservation of Green Peafowl (Pavo muticus) in China"

_sustainability, doi:10.3390/su132111723_

Round 1

Reviewer 1 Report

Yang et al. presented an interesting and important study on the reduction of the ecological habitat of the Chinese green peacock (Pavo muticus) in Yunnan Province, China, and the potential measurements that can be applied to protect it in the future.

This study took many years to collect ancient Chinese documents and on-site interview records to speculated on the possible locations of Chinese green peacocks in the past. Although only a three-month survey using camera traps, it is still of reference value for the conservation of Chinese green peacocks. However, as to the current environmental conditions have an impact on the conservation of Chinese green peacocks or potential approaches that can be applied, only some practical suggestions are mentioned. Some references are suggested to be cited to strengthen this article's description on the reduction of species richness caused by human pressure, and to propose conservation strategies. For example, in the discussion part, the author speculates that hunting, trade, and habitat degradation have caused the reduction of the Chinese green peafowl's range and population decline. However, it is also possible that urbanization or the increase of arable land may cause environmental fragmentation can also make species conservation difficult. It is recommended to add literatures of description on other species, urbanization and environmental fragmentation, such as Oryx, 2015, 49(2) pp. 261-269; Org. Agr., 2020, 10:409–418; and Biol. Conserv., 2016, 195:264–271. to increase persuasiveness and attract readers who are interested in the subject of species abundance and environmental conservation.

Author Response

Author response: We appreciate your valuable suggestions. In the Discussion section, we mentioned that immigration, human population growth, hunting, trade, and habitat degradation are important reasons for range retraction and population decline of Green Peafowl in historical times in China. We also agree that urbanization and increase of arable land have strong negative effects on Green Peafowl persistence. Therefore, we have added a paragraph in the beginning of the Introduction section to present the general background. For many species, the severe populations decline and distributions shrinkage caused by human activities have also been mentioned in the paragraph:

“Anthropogenic disturbances have threatened species persistence [1]. Many species have experienced severe declines in distribution and the populations have been restricted to fragmented habitats or small protected areas [2]. Understanding long-term species distribution change and current distribution pattern is crucial to develop appropriate conservation strategies for threatened species [3].”

Reviewer 2 Report

I really enjoyed this paper combining historical records, camera traps and maxent modelling to understand the changes in the range of green peafowl in China, and suggesting plans for its conservation. I suspect I might be reviewing a later version of this paper, as the narrative is excellent, the research questions clear and the modelling done very well. I have some small comments.

Paragraph 1: I would prefer to see a broader opening that mirrors the conclusion of the paper - instead of going straight into the study species, the reader should get context about the current state of biodiversity in China, and how similar studies of historical records have been used in conservation studies. Then in paragraph three you can state that such is the case for Green Peafowl - this would attract more readers to this article as well. 

Paragraph 2 needs a bit of introduction to the camera trap and Maxent method - I find these methods appropriate here, but I feel you need some introductory context to say why they are the ideal means to answer the questions. As it stands of the three research questions, question 2 is not supported by the introduction.

Methods section 2.2.1 - it would be useful to have more information on how the interviews were conducted - as it stands, it feels a bit haphazard and random.

in 2.2.1 you mention eggs and nestlings - and I guess these were on transect surveys - maybe this is in the wrong paragraph and should belong in the section on line transects (as it is not mentioned there that eggs and nestling were recorded)

I think although relatively simple the Maxent methods are appropriate for this study - I know there is a lot of critique of this method nowadays, but it is suitable here especially with the combination of historic and accurately collected records.

Figure 1 is so interesting!

Page 9 paragraph two - Green peafowl WAS distributed widely...

Paragraph three - at a much finer scale

Conservation section - I think before reintroduction is mentioned, you need to bring up law enforcement and if possible give an example of where this worked before - because reintroduction will fail if law enforcement is not there first - there is also a strange font change here on my version.

Overall I applaud the authors for this work.
